# AvatarGO: Zero-shot 4D Human-Object Interaction Generation and Animation

**Yukang Cao**[1*] **Liang Pan**[2†‡] **Kai Han**[3] **Kwan-Yee K. Wong**[3] **Ziwei Liu**[1†]

[1]S-Lab, Nanyang Technological University, [2]Shanghai AI Laboratory, [3]The University of Hong Kong

https://yukangcao.github.io/AvatarGO/

## Abstract

Recent advancements in diffusion models have led to significant improvements in the generation and animation of 4D full-body human-object interactions (HOI). Nevertheless, existing methods primarily focus on SMPL-based motion generation, which is limited by the scarcity of realistic large-scale interaction data. This constraint affects their ability to create everyday HOI scenes. This paper addresses this challenge using a zero-shot approach with a pre-trained diffusion model. Despite this potential, achieving our goals is difficult due to the diffusion model's lack of understanding of *"where"* and *"how"* objects interact with the human body. To tackle these issues, we introduce **AvatarGO**, a novel framework designed to generate animatable 4D HOI scenes directly from textual inputs. Specifically, **1)** for the *"where"* challenge, we propose **LLM-guided contact retargeting**, which employs Lang-SAM to identify the contact body part from text prompts, ensuring precise representation of human-object spatial relations. **2)** For the *"how"* challenge, we introduce **correspondence-aware motion optimization** that constructs motion fields for both human and object models using the linear blend skinning function from SMPL-X. Our framework not only generates coherent compositional motions, but also exhibits greater robustness in handling penetration issues. Extensive experiments with existing methods validate AvatarGO's superior generation and animation capabilities on a variety of human-object pairs and diverse poses. As the first attempt to synthesize 4D avatars with object interactions, we hope AvatarGO could open new doors for human-centric 4D content creation.

## 1 Introduction

The creation of 4D human-object interaction (HOI) holds immense significance across a wide range of industries, including augmented/virtual reality (AR/VR) and game development, as it forms the foundation of the 4D virtual world. Traditionally, developing such models has required extensive human effort and specialized engineering expertise. Fortunately, thanks to the collections of HOI datasets (Li et al., 2023b; Bhatnagar et al., 2022; Jiang et al., 2023a) and the recent advancements in diffusion models (Saharia et al., 2022; Ramesh et al., 2022; Balaji et al., 2022; Stability.AI, 2022; 2023), existing HOI generative techniques (Zhang et al., 2022; 2023; 2024; Shafir et al., 2023; Kapon et al., 2024; Chen et al., 2024a) have exhibited promising capabilities by generating 4D human motions with object interactions from textual inputs. Nonetheless, these methods primarily focus on SMPL-based (Loper et al., 2015; Pavlakos et al., 2019) motion generation, which struggles to capture the realistic appearance of subjects encountered in everyday life. Although InterDreamer (Xu et al., 2024b) has recently proposed to generate text-aligned 4D HOI sequences in a zero-shot manner, their output is still largely constrained by the SMPL model. This highlights a pressing need for more realistic and generalizable methods tailored specifically to model 4D human-object interactive content. We take the initiative and showcase the potential of addressing this challenge by leveraging the 3D generative methods in a zero-shot manner.

In recent times, 3D generative methods (Poole et al., 2022; Tang et al., 2023; Liu et al., 2023c; Lin et al., 2023; Wang et al., 2023e; Cao et al., 2023b; Liao et al., 2023) and Large Language

---

*Part of the work has been done when interning at Shanghai AI Laboratory. † Corresponding authors
‡ Project lead

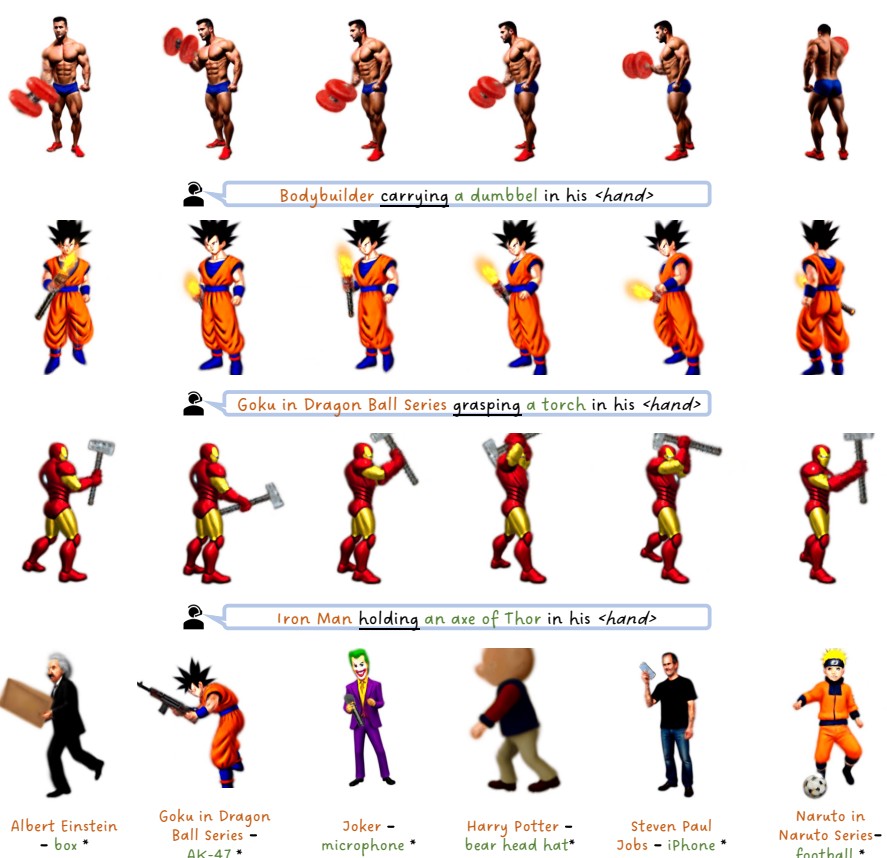

Figure 1: **Examples of 4D animation results obtained via AvatarGO.** AvatarGO effectively produces diverse human-object compositions with correct spatial correlations and contact areas. It achieves joint animation of humans and objects while avoiding penetration issues.

Models (LLMs) (Wu et al., 2023a) have garnered increasing interest. These progressives have led to the development of text-guided 3D compositional generation techniques that are capable of comprehending intricate relations and creating complex 3D scenes incorporating multiple subjects. Notably, GraphDreamer (Gao et al., 2023) utilizes LLMs to construct a graph where nodes represent objects and edges denote their relations. ComboVerse (Chen et al., 2024b) proposes spatial-aware score distillation sampling to amplify the spatial correlation. Subsequent studies (Epstein et al., 2024; Zhou et al., 2024) further explore the potential of jointly optimizing layouts to composite different components.

Despite the promising performance demonstrated by existing methods, they encounter two major challenges in generating 4D HOI scenes: *1) Incorrect contact area:* While LLMs excel at capturing the relationships, optimization with diffusion models faces difficulties in accurately defining the contact area between various objects, particularly those with complex articulated structures like human bodies. Although efforts like InterFusion (Dai et al., 2024) have constructed 2D human-object interaction datasets to retrieve human poses from text prompts, they still encounter challenges in defining the optimal contact body parts for cases outside the training distribution. *2) Limitations in 4D compositional animation:* While existing techniques like DreamGaussian4D (Ren et al., 2023) and TC4D (Bahmani et al., 2024) employ video diffusion models (Blattmann et al., 2023; Guo et al., 2023) to animate 3D static scenes, they often treat the entire scene as one subject during optimization, leading to unrealistic animation results. Despite initiatives like Comp4D (Xu et al., 2024a), which utilize trajectories to animate 3D objects individually, modeling contact between various subjects remains a challenge.

In this paper, we propose **AvatarGO**, a novel framework for compositional 4D avatar generation with object interactions. By taking the text prompts as inputs, we assume that the 3D human and object models as well as the human motion sequences can be individually generated by adopting existing

generative techniques (Tang et al., 2023; Liu et al., 2023d; Zhang et al., 2023; 2024). Specifically, we adopt DreamGaussian4D (Ren et al., 2023) as our baseline considering its superior training efficiency and focus on addressing the challenges associated with human-object interactions. To achieve this objective, AvatarGO integrates two key innovations to learn *"where"* and *"how"* the object should interact with the human body: **1) LLM-guided contact retargeting.** Given the limited availability of human-object interaction images in the 2D dataset used for diffusion model training, it's difficult to identify the most appropriate contact area between humans and objects. To tackle this issue, we propose leveraging Lang-SAM (lan, 2023) to identify the contact body part from text prompts, which serves as the initialization for the optimization procedure. **2) Correspondence-aware motion optimization.** Building upon the observation that penetration is absent in static composited models, we introduce correspondence-aware motion optimization that leverages SMPL-X as an intermediary to maintain the correspondence between humans and objects when they are animated to a new pose, thus demonstrating greater robustness in handling penetration issues.

We thoroughly assess AvatarGO by compositing diverse pairs of 3D humans and objects and animating them across various motion sequences (see Fig. 1). Our experimental results show that our method excels at identifying optimal contact areas and exhibits greater robustness in handling penetration issues during animation, significantly outperforming existing techniques. We will make our code publicly available.

## 2 RELATED WORK

**3D Content Generation.** Leveraging advances in diffusion-based text-to-2D image generation (Saharia et al., 2022; Ramesh et al., 2022; Balaji et al., 2022; Stability.AI, 2022; 2023), DreamFusion introduced Score Distillation Sampling (SDS) to generate 3D content via pre-trained models, utilizing technologies like NeRF (Mildenhall et al., 2020), DMTET (Shen et al., 2021), 3D Gaussian Splatting (Kerbl et al., 2023)). Subsequent research has focused on enhancing output quality (Lin et al., 2023; Chen et al., 2023b; Wang et al., 2023e), controlling generation processes (Metzer et al., 2022; Seo et al., 2023), improving training efficiency (Wang et al., 2023a; Wu et al., 2024; Tang et al., 2023), and extending capabilities on 3D texturing (Richardson et al., 2023; Cao et al., 2023a; Chen et al., 2023a; Tang et al., 2024b). Addressing 3D human body complexity, recent studies (Cao et al., 2023b;c; Han et al., 2023; Liao et al., 2023; Jiang et al., 2023b; Huang et al., 2023b; Kolotouros et al., 2023; Zeng et al., 2023; Huang et al., 2023a; Cao et al., 2024) have been proposed for creating controllable 3D human avatars, although these still require significant input-specific training time. The proliferation of large 3D datasets (Deitke et al., 2023; 2024; Wu et al., 2023b) has propelled 3D generation techniques forward. Notably, Zero-1-to-3 (Liu et al., 2023c), Zero123++ (Shi et al., 2023a), and MVDream (Shi et al., 2023b) use 2D diffusion models to generate consistent multi-view images, serving as inputs for efficient 3D model generation tools like SyncDreamer (Liu et al., 2023e), Wonder3D (Long et al., 2023), One-2-3-45 (Liu et al., 2023b;a), UniDream (Liu et al., 2023f), MVDiffusion++ (Tang et al., 2024c), and Make-Your-3D (Liu et al., 2024a). Additionally, building on transformer (Vaswani et al., 2017) and image processor advancements (e.g., DINO (Caron et al., 2021; Oquab et al., 2023)), Large Reconstruction Models (Hong et al., 2023; Wang et al., 2023b; Xu et al., 2023; Li et al., 2023a) implement transformer-based architectures to derive 3D tri-plane tokens from image features. 3DTopia (Hong et al., 2024) uses hybrid diffusion priors to produce high-fidelity 3D objects. Meanwhile, methods like LGM (Tang et al., 2024a), CRM (Wang et al., 2024), and GRM (Yinghao et al., 2024) explore various 3D representations for improved performance, such as 3D Gaussian Splatting (Kerbl et al., 2023) and FlexiCube (Shen et al., 2023). Despite these advances, challenges remain in generating complex compositional 3D scenes.

**3D Compositional Generation.** To address the compositional nature of 3D content, a few efforts have been made recently. *Epstein et al* (Epstein et al., 2024) and GALA3D (Zhou et al., 2024) propose optimizing component layouts for integrated object scenes. ComboVerse (Chen et al., 2024b) introduces spatial-aware score distillation sampling (SSDS) to effectively learn object spatial relations. GraphDreamer (Gao et al., 2023) uses large language models to form graph structures where nodes and edges represent objects and their relationships, respectively, showing promising results. Challenges remain in modeling interactions between humans and objects. InterFusion (Dai et al., 2024) develops a 2D dataset for human-object interactions, enabling text-guided pose retrieval and scene generation. However, this approach lacks precise control over interaction areas and is not readily adaptable to 4D scenarios.

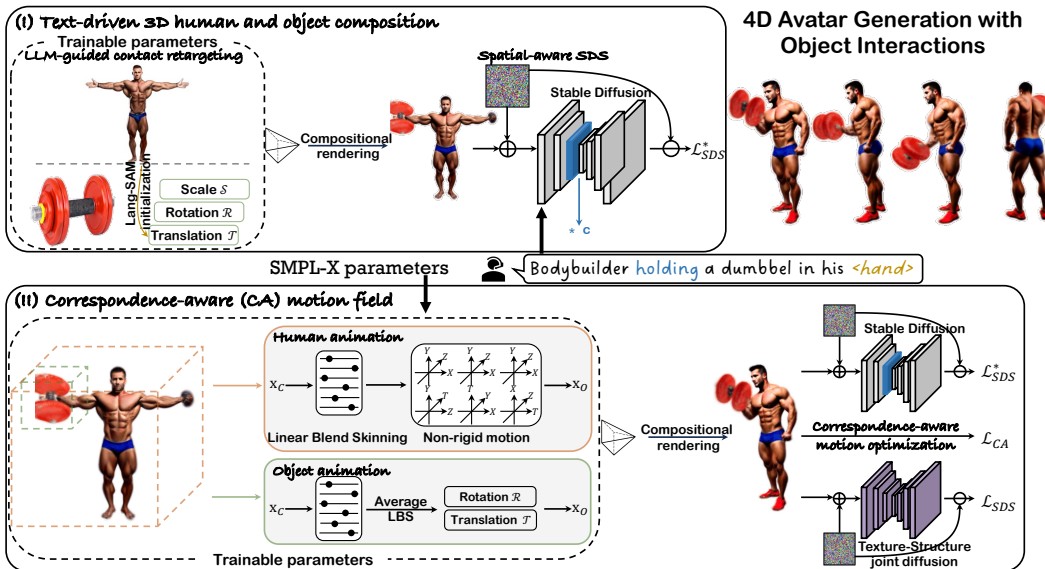

Figure 2: **Overview of AvatarGO.** AvatarGO takes the text prompts as input to generate 4D avatars with object interactions. At the core of our network are: 1) *Text-driven 3D human and object composition* that employs large language models to retarget the contact areas from texts and spatial-aware SDS to composite the 3D models. 2) *Correspondence-aware motion optimization* which jointly optimizes the animation for humans and objects. It effectively maintains the spatial correspondence during animation, addressing the penetration issues.

**4D Content Generation.** Recent advances in video diffusion models and score distillation sampling have spurred a variety of 4D scene generation techniques. Make-A-Video3D (MAV3D) (Singer et al., 2023) utilizes HexPlane features for 4D representations. 4D-fy (Bahmani et al., 2023) and DreamGaussian4D (Ren et al., 2023) employ multi-stage optimization pipelines to transform static 3D into dynamic 4D scenes. Dream-in-4D (Zheng et al., 2023) allows for personalized 4D generation using image guidance, while Consistent4D (Jiang et al., 2023c) uses video inputs with RIFE (Huang et al., 2022) and a super-resolution module for scene creation. 4DGen (Yin et al., 2023) and AnimatableDreamer (Wang et al., 2023c) focus on controllable motion generation via driving videos. More recently, Comp4D (Xu et al., 2024a) and TC4D (Bahmani et al., 2024) introduced trajectory-based approaches for creating 4D compositional scenes. While these technologies show promise, they often struggle to produce 4D avatars that effectively interact with objects. Although GAvatar (Yuan et al., 2023) excels in 4D human animation, its object interaction capabilities are limited. On the other hand, SMPL-based 4D HOI generation remains an open challenge for 4D content generation. Methods like Chain-of-Contacts Xiao et al. (2023), CG-HOI Diller & Dai (2024), and others Liu et al. (2024b); Jiang et al. (2024); Wu et al. (2022); Jiang et al. (2023a); Yang et al. (2024); Wang et al. (2023d) have significantly advanced the field. However, these techniques primarily focus on generating motion sequences for SMPL models, which lack intricate clothing details. Additionally, these methods rely on training datasets, limiting their adaptability in practical scenarios.

## 3 METHODOLOGY

Given a generated 3D avatar and a specific 3D object, AvatarGO generates compositional 4D avatars with object interactions based on text instructions. In the subsequent sections, we first introduce the preliminaries (in Sec. 3.1), including static 3D content generation and parametric human model SMPL-X. Next, we will describe the key components of AvatarGO, including (1) text-driven 3D human and object composition (in Sec. 3.2), and (2) correspondence-aware motion optimization for achieving synchronized human and object animation (in Sec. 3.3). The overview of AvatarGO is shown in Fig. 2.

### 3.1 PRELIMINARIES

**3D Model Generation.** Recently, DreamGaussian (Tang et al., 2023) showcases promising results with largely improved training efficiency by incorporating two major components:

(1) *3D Gaussian Splatting (3DGS).* 3DGS (Kerbl et al., 2023) directly defines the 3D space through a set of Gaussians parameterized by their 3D position $\mu$, opacity $\alpha$, anisotropic covariance $\Sigma$, and spherical harmonic coefficients $sh$. The $sh$ term is used to capture the view-dependent appearance of the scene and $\Sigma$ can be decomposed to:

$$\Sigma = RSS^T R^T, \tag{1}$$

where $R$ is the rotation matrix expressed by a quaternion $q \in \mathbf{SO}(3)$, and $S$ is the scaling matrix, represented by a 3D vector $s$. Essentially, each Gaussian centered at point (mean) $\mu$ is defined as:

$$G(\mathbf{x}, \mu) = e^{-\frac{1}{2}(\mathbf{x}-\mu)^T \Sigma^{-1}(\mathbf{x}-\mu)}, \tag{2}$$

where $\mathbf{x}$ is the 3D query point.

For rendering the 3D Gaussians onto the 2D image space, 3DGS incorporates a tile-based rasterizer and point-based $\alpha$-blend rendering. Specifically, the color $C(u)$ of a pixel $u$ can be calculated as:

$$C(u) = \sum_{i \in N} T_i c_i \alpha_i \mathcal{SH}(sh_i, v), \quad T_i = G(\mathbf{x}, \mu_i) \prod_{j=1}^{i-1} (1 - \alpha_j G(\mathbf{x}, \mu_j)), \tag{3}$$

where $T$ represents the transmittance, $\mathcal{SH}$ denotes the spherical harmonic function, and $v$ indicates the viewing direction. By optimizing the Gaussian attributes $\{G : \mu, q, s, \sigma, c\}$ and dynamically adjusting the density of 3D Gaussians (*i.e.*, densifying and pruning), DreamGaussian achieves high-quality generations from either textual or visual inputs.

(2) *Score Distillation Sampling (SDS).* Starting with the latent feature $z$ extracted from a 3DGS rendering $x$, SDS introduces random noise $\epsilon$ to $z$, yielding a noisy latent variable $z_t$. This variable is then processed by a pre-trained denoising function $\epsilon_\phi(z_t; y, t)$ to estimate the added noise. The SDS loss then calculates the difference between predicted and added noise, with its gradient calculated by:

$$\nabla_\theta \mathcal{L}_{\text{SDS}}(\phi, g(\theta)) = \mathbb{E}_{t, \epsilon \sim \mathcal{N}(0,1)} \left[ w(t) \left( \epsilon_\phi(z_t; y, t) - \epsilon \right) \frac{\partial z}{\partial x} \frac{\partial x}{\partial \theta} \right], \tag{4}$$

where $y$ denotes the text embedding, $w(t)$ weights the loss from noise level $t$. We do not apply the mesh extraction and texture optimization proposed in DreamGaussian to obtain the 3D models.

**SMPL-X** (Loper et al., 2015; Pavlakos et al., 2019). With pose parameter $\theta$, shape parameter $\beta$, and expression parameter $\phi$ as inputs, SMPL-X maps the canonical model to the observation space:

$$M(\beta, \theta, \phi) = \text{LBS}(\boldsymbol{T}(\beta, \theta, \phi), J(\beta), \theta, \mathcal{W}), \tag{5a}$$

$$\boldsymbol{T}(\beta, \theta, \phi) = \mathbf{T} + B_s(\beta) + B_e(\phi) + B_p(\theta), \tag{5b}$$

where $M$ denotes the function defining the mesh model of a human body, and $\boldsymbol{T}$ represents the transformed vertices. $\mathcal{W}$ stands for blend weights, $B_s$, $B_e$, and $B_p$ are functions respectively for shape, expression, and pose blend shapes. $\text{LBS}(\cdot)$ indicates the linear blend skinning function that poses each body vertex of SMPL-X according to:

$$\mathbf{v}_o = \mathcal{G} \cdot \mathbf{v}_c, \quad \mathcal{G} = \sum_{k=1}^{K} w_k \mathcal{G}_k(\theta, j_k), \tag{6}$$

where $\mathbf{v}_c$ and $\mathbf{v}_o$ represent SMPL-X vertices under the canonical pose and observation space, respectively. $w_k$ is the skinning weight, $\mathcal{G}_k(\theta, j_k)$ is the affine deformation that maps the $k$-th joint $j_k$ from the canonical space to observation space, and $K$ denotes the number of neighboring joints.

## 3.2 TEXT-DRIVEN 3D HUMAN AND OBJECT COMPOSITION

With the help of DreamGaussian (Tang et al., 2023), we efficiently generate the 3D avatar $G_h$ and the 3D object $G_o$ individually based on 3DGS and SDS (discussed in Sec. 3.1). We noticed that even with manual adjustments, such as rescaling and rotating the 3D objects, it's difficult to directly rig the generated 3D human and object models accurately (see Appendix). Therefore, we strive to seamlessly composite $G_h$ and $G_o$ based on the text prompt in this stage. Specifically, the Gaussian attributes of $G_h$ and $G_o$ would be optimized, as well as three trainable global parameters of $G_o$, including rotation $\mathcal{R} \in \mathbb{R}^4$, scaling factor $\mathcal{S} \in \mathbb{R}$, and the translation matrix $\mathcal{T} \in \mathbb{R}^3$:

$$\mathbf{X}_{G_o} := \mathcal{S} \cdot (\mathbf{X}_{G_o} \cdot \mathcal{R} + \mathcal{T}), \tag{7}$$

where $\mathbf{X}_{G_o}$ is the set of static Gaussian points.

However, solely utilizing SDS for optimization could frequently lead to disproportionate relationships and erroneous contact areas (see Fig. 3). This issue can be attributed to two potential factors: (1) the absence of emphasis on words describing human-object interaction, which decreases the model's ability to comprehend the relationships between humans and objects; (2) the complexity inherent in human subjects, posing challenges for the diffusion model to identify the most suitable contact areas (see Sec. 4.3).

**Spatial-aware SDS (SSDS).** Following ComboVerse (Chen et al., 2024b), we employ SSDS to facilitate the compositional 3D generation between the human and the object. Specifically, SSDS augments the SDS with a spatial relationship between the human and the object by scaling the attention maps of the designated tokens <token*> with a constant factor $c$ (where $c > 1$):

$$\text{ATT} := \begin{cases} c \cdot \text{ATT}_{<\texttt{token}>}, & \text{if} \quad <\texttt{token}> = <\texttt{token}^*>, \\ \text{ATT}_{<\texttt{token}>}, & \text{otherwise.} \end{cases} \tag{8}$$

Here, <token*> corresponds to the tokens encoding the human-object interaction term, such as <'holding'>, which can be identified through Large Language Models (LLMs) or specified by the user. Consequently, the spatial-aware SDS loss can be written as:

$$\nabla_\theta \mathcal{L}_{\text{SSDS}}(\phi^*, g(\theta)) = \mathbb{E}_{t,\epsilon \sim \mathcal{N}(0,1)} \left[ w(t) \left( \epsilon_{\phi^*}\left( \boldsymbol{z}_t; y, t \right) - \epsilon \right) \frac{\partial \boldsymbol{z}}{\partial \boldsymbol{x}} \frac{\partial \boldsymbol{x}}{\partial \theta} \right], \tag{9}$$

where $\phi^*$ denotes the pre-trained denoising function with the adjusted attention maps.

**LLM-guided Contact Retargeting.** While spatial-aware SDS could benefit in understanding spatial correlations, it still faces difficulties in identifying the most appropriate contact area (See Fig. 3), which serves as a key component for human-object interaction. According to our studies (see Appendix for visualization), the diffusion model struggles to accurately estimate contacts, even in the 2D images generated for human-object interaction. To tackle this issue, we propose leveraging Lang-SAM (lan, 2023) to identify the contact area from text prompts. Specifically, starting from the 3D human model $G_h$, we render it from a frontal viewpoint to produce the image $I$. This image, alongside textual inputs, undergoes Lang-SAM model to derive 2D segmentation masks $\mathcal{M}$:

$$\text{LangSAM}(I, <\texttt{body-part}>) \rightarrow \mathcal{M}, \tag{10}$$

where <body-part> represents the text describing the human body part, such as <'hand'>. Subsequently, we back-project the 2D segmentation labels onto the 3D Gaussians via inverse rendering (Chen et al., 2023c). Specifically, for each pixel $u$ on the segmentation maps, we update the mask value (0 or 1) back to the Gaussians via:

$$w_i = \sum_{i \in \mathcal{N}} o_i(u) \times T_i(u) \times \mathcal{M}(u), \tag{11}$$

where $w_i$ represents the weight of the $i$-th Gaussian, $\mathcal{N}$ is the collection of Gaussians that can be projected onto the pixel $u$. $o(\cdot)$, $T(\cdot)$, and $\mathcal{M}(\cdot)$ respectively denote the opacity, transmittance, and segmentation mask value. Following the weight updates, we assess whether a Gaussian corresponds to the segmented region of the human body part by comparing its weight against a pre-defined threshold $a$. We then initialize the translation parameter $\mathcal{T}$ according to:

$$\mathcal{T} = (\boldsymbol{w}^T * \boldsymbol{\mu}) / \sum \boldsymbol{w}, \tag{12}$$

where $\boldsymbol{w} = \{w_1, ..., w_N | w_i = 0/1\} \in \mathbb{R}^{N \times 1}$, $\boldsymbol{\mu} = \{\mu_1, ..., \mu_N\} \in \mathbb{R}^{N \times 3}$, and $N$ is the number of Gaussain points within the human model $G_h$.

### 3.3 Correspondence-aware Motion Field

Following the compositional integration of 3D humans and objects, animating them synchronously presents an additional challenge owing to potential penetration issues. This problem stems from the absence of a well-defined motion field for the object. To this end, we establish the motion fields for both human and object models using the linear blend skinning function from SMPL-X (as in Eq. 6), and propose a correspondence-aware motion optimization aimed at optimizing the trainable global

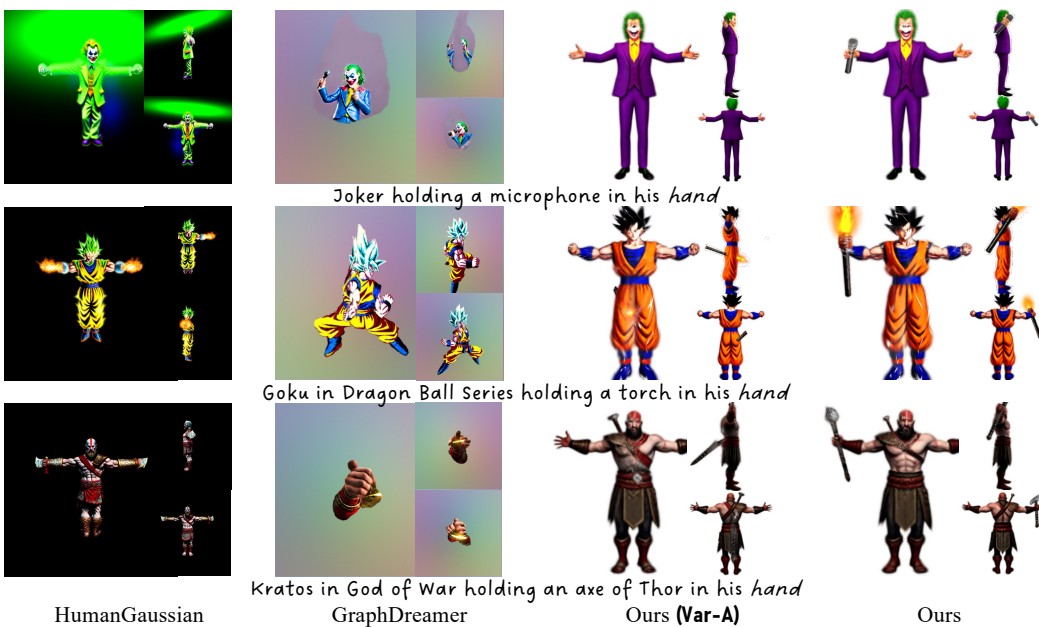

Figure 3: **Comparisons on 3D compositional generations.**

parameters of the object model, *i.e.*, rotation ($\mathcal{R}$) and translation ($\mathcal{T}$), to improve robustness against penetration issues between humans and objects.

**Human Animation.** Given the motion sequence, we first construct a deformation field, which consists of two components: (1) articulated deformation utilizing the SMPL-X linear blend skinning function $\text{LBS}(\cdot)$, and (2) non-rigid motion learning the offset based on HexPlane features (Cao & Johnson, 2023), to deform the point $\mathbf{x}_c$ from the canonical space to $\mathbf{x}_o$ in the observation space:

$$\mathbf{x}_o = \mathcal{G} \cdot \mathbf{x}_c + \text{MLP}(F(\mathbf{x}_c, \mathbf{t})), \tag{13}$$

where $F(\cdot)$ denotes the HexPlane-based feature extraction network, and $\mathbf{t}$ indicates the timestamp. We derive $\mathcal{G}$ from the closet canonical SMPL-X vertex to $\mathbf{x}_c$ via Eq. 6.

**Object Animation.** Similar to the human animation, we calculate the deformation matrix $\mathcal{G}_c$ for each Gaussian point $\mathbf{x}$ within the object model $G_o$ based on its closest canonical SMPL-X vertex. Given our experimental definition of 3D objects as rigid bodies, we then compute their average to establish the intermediate motion field for the object:

$$\mathbf{X}_o = \mathcal{G}'_c \cdot \mathbf{X}_c, \quad \mathcal{G}'_c = \frac{\sum_{i \in [1, M]} \mathcal{G}_{c_i}}{M}, \tag{14}$$

where $\mathbf{X}_o = \{\mathbf{x}_{o_1}, ..., \mathbf{x}_{o_M}\}$, $\mathbf{X}_c = \{\mathbf{x}_{c_1}, ..., \mathbf{x}_{c_M}\}$, and $M$ is the total number of Gaussian points within $G_o$. Although animating the object directly using SMPL-X linear blend skinning may seem like a simple solution, it can result in penetration issues between the human and the object (see Fig. 6). This challenge arises primarily from the absence of proper constraints to maintain the correspondence between these two models.

**Correspondence-aware Motion Optimization.** Drawing insight from the fact that our method is robust in handling penetration issues in static composited models across various scenarios, we propose a correspondence-aware motion optimization to preserve the correspondence between human and object, thereby addressing the penetration problem. Specifically, we extend the above motion field (Eq. 14) to include two additional trainable parameters $\mathcal{R}$ and $\mathcal{T}$:

$$\mathbf{X}_o := \mathbf{X}_o \cdot \mathcal{R} + \mathcal{T}. \tag{15}$$

where $\mathbf{X}_o$ is obtained in Eq. 14. Rather than naïvely optimizing the parameters via SDS, we propose a novel correspondence-aware training objective that leverages SMPL-X as an intermediary to maintain the correspondence between human and object when they are animated to a new pose:

$$\mathcal{L}_{CA} = \text{MSE}(\mathcal{G}_{\mathbf{c}}, \mathcal{G}_{\mathbf{o}}), \quad \mathcal{G}_{\mathbf{c}} = \{\mathcal{G}_{c_0}, ..., \mathcal{G}_{c_M}\}, \quad \mathcal{G}_{\mathbf{o}} = \{\mathcal{G}_{o_0}, ..., \mathcal{G}_{o_M}\} \tag{16}$$

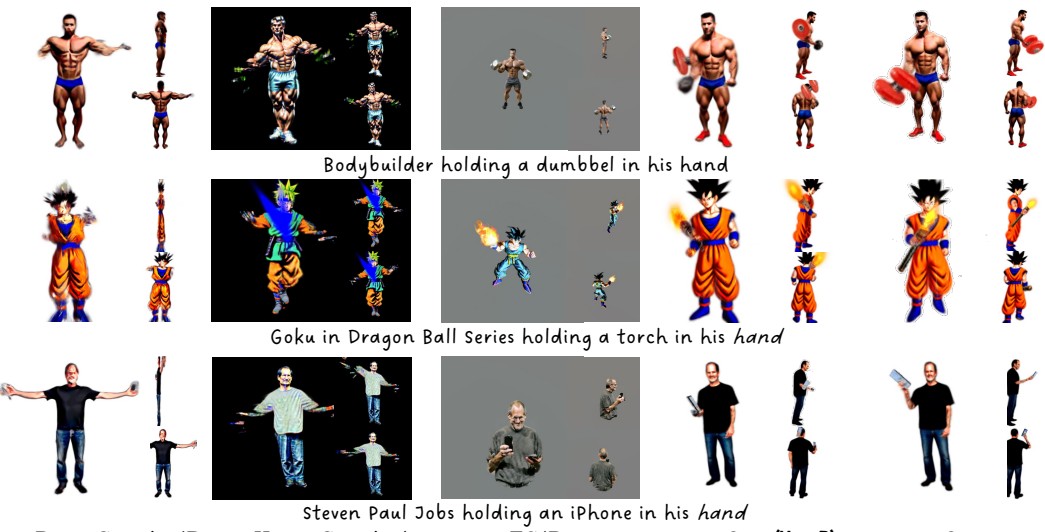

DreamGaussian4D    HumanGaussian*    TC4D    Ours **(Var-B)**    Ours

Figure 4: **Comparisons on 4D avatar animation with object interactions.** We present HumanGaussian in three poses, whereas other methods are shown in three poses and three different views. '∗' indicates that the SMPL LBS function is directly employed on HumanGaussian directly for animation, without additional training as in other methods.

where $\mathcal{G}_{c_i}$ is derived via Eq. 6 by finding the nearest vertex in the canonical SMPL model to $\mathbf{x}_{c_i}$, while $\mathcal{G}_{o_i}$ is similarly obtained by locating the closest vertex in the observed SMPL model to $\mathbf{x}_{o_i}$.

In addition to our correspondence-aware loss, we also incorporate the spatial-aware SDS as in Eq. 9 and the texture-structure joint SDS from HumanGaussian (Liu et al., 2023d) to enhance the overall quality:

$$\nabla_\theta \mathcal{L}_{\mathrm{SDS}}(\phi, g(\theta)) = \lambda_1 \cdot \mathbb{E}_{t,\epsilon \sim \mathcal{N}(0,1)} \left[ w(t) \left( \epsilon_\phi \left( \boldsymbol{z}_{x_t}; y, t \right) - \epsilon_{\boldsymbol{x}} \right) \frac{\partial \boldsymbol{z_x}}{\partial \boldsymbol{x}} \frac{\partial \boldsymbol{x}}{\partial \theta} \right)$$

$$+ \lambda_2 \cdot \mathbb{E}_{t,\epsilon \sim \mathcal{N}(0,1)} \left[ w(t) \left( \epsilon_\phi \left( \boldsymbol{z}_{d_t}; y, t \right) - \epsilon_{\boldsymbol{d}} \right) \frac{\partial \boldsymbol{z_d}}{\partial \boldsymbol{d}} \frac{\partial \boldsymbol{d}}{\partial \theta} \right], \qquad (17)$$

where $\lambda_1$ and $\lambda_2$ are hyper-parameters to balance the impact of structural and textural losses, while $\boldsymbol{d}$ denotes the depth renderings.

The overall loss function to optimize the 4D animative scene is then given by:

$$\mathcal{L} = \lambda_{CA} \cdot \mathcal{L}_{CA} + \lambda_{\mathrm{SDS}} \cdot \mathcal{L}_{\mathrm{SDS}} + \lambda_{\mathrm{SSDS}} \cdot \mathcal{L}_{\mathrm{SSDS}}, \qquad (18)$$

where $\lambda_{CA}$, $\lambda_{\mathrm{SDS}}$, and $\lambda_{\mathrm{SSDS}}$ represents weights to balance the respective losses.

## 4 EXPERIMENTS

We now validate the effectiveness and capability of our proposed framework to animate various 3D avatar-object pairs with different poses and provide comparisons with existing 3D and 4D compositional generation methods.

**Implementation Details.** We follow DreamGaussian4D (Ren et al., 2023) to implement the 3D Gaussian Splatting (Kerbl et al., 2023) and the HexPlane (Cao & Johnson, 2023) in our method. We utilize the pre-trained Texture-Structure joint diffusion model from HumanGaussian (Liu et al., 2023d) and version 2.1 of Stable Diffusion (Stability.AI, 2022) to respectively calculate the SDS and spatial-aware SDS in our implementation. Typically, for each 3D avatar-object pair, we train the 3D stage with a batch size of 16 for 400 epochs, and the 4D stage with a batch size of 10 for 400 epochs. The training takes around 10 minutes for the 3D stage and 20 minutes for the 4D stage on a single NVIDIA A100 GPU. We use Adam (Kingma & Ba, 2015) optimizer for back-propagation. Additional implementation details can be found in the Appendix.

**Comparison Methods for 3D Static Generation.** We first compare the 3D static generation results with HumanGaussian (Liu et al., 2023d) and GraphDreamer (Gao et al., 2023). Since

ComboVerse (Chen et al., 2024b) lacks an official code release and relies on image inputs, we compare static AvatarGO with an alternative variant, *i.e.*, "Ours (**Var-A**)", by only using the spatial-aware score distillation sampling (SSDS) in ComboVerse to composite 3D humans and avatars. We cannot compare with GALA3D as their source code is not publicly accessible.

**Comparison Methods for 4D Animation.** Since there are no specific methods tailored for 4D avatar animation with object interactions, we access AvatarGO's efficacy against three recent 4D generation techniques (i.e., DreamGaussian4D (Ren et al., 2023), HumanGaussian (Liu et al., 2023d), and TC4D (Bahmani et al., 2024)), as well as one alternative variant "Ours (**Var-B**)". To implement **Var-B**, we utilize human hand motion sequences as trajectories to guide the transformation of 3D objects and follow Comp4D to integrate the video diffusion model to compute SDS. Because InterDreamer (Xu et al., 2024b) and InterFusion (Dai et al., 2024) have not released their code, we could not include their results for comparison. See more motivation for designing "Ours (Var-A)" and "Ours (Var-B)" in Appendix.

## 4.1 QUALITATIVE EVALUATIONS

**4D Avatar Generation with Object Interaction.** In Fig. 1, we present a diverse collection of avatar-object pairs that are animated to different poses. These renderings consistently showcase high-fidelity results from various viewpoints. Thanks to our proposed LLM-guided contact retargeting and correspondence-aware motion optimization, our method can deliver appropriate human-object interactions and demonstrate superior robustness to the penetration issues.

**Comparison on 3D Generation.** We provide qualitative comparisons with existing methods on 3D generation in Fig. 3. We can observe: 1) without the aid of LLMs, HumanGaussian struggles to determine the spatial correlations between humans and objects; 2) Despite using graphs to establish relationships, GraphDreamer is confused by the meaningful contact, resulting in unsatisfactory outcomes. 3) Optimizing $\mathcal{R}$, $\mathcal{S}$, and $\mathcal{T}$ with only SSDS is inadequate to move the object to the correct area. Conversely, AvatarGO consistently outperforms with precise human-object interactions.

**Comparisons on 4D Animation.** In Fig. 4, we compare our 4D animation results with SOTA methods. We take the rendering from our 3D compositions stage as the input for DreamGaussian4D. The following observations can be made: 1) Even with human-object interaction images, Dream-Gaussian4D, which employs video diffusion models, struggles with animating the composited scene. 2) Direct animation via SMPL LBS function, as in HumanGaussian, tends to yield unsmooth results, especially for the arms. 3) TC4D faces similar issues as the DreamGaussian4D. Meanwhile, it treats the entire scene as a single entity, lacking both local and large-scale motions for individual objects. 4) One may think applying trajectory to objects seems like a simple solution (as in Comp4D). However, as seen in "Ours (**Var-B**)", it can disrupt spatial correlations between humans and objects. These points further validate the necessity of AvatarGO. Our method can consistently deliver superior results with correct relationships and better robustness to penetration issues See the Appendix for more comparisons.

## 4.2 QUANTITATIVE EVALUATIONS

**CLIP-based Metrics.** We use CLIP-based metrics (CLIP-Score (CLIP-S), CLIP image similarity (CLIP-Image), and CLIP Directional Similarity (CLIP-DS) (Brooks et al., 2023; Gal et al., 2022)) with CLIP-ViT-L/14 model. Among

Table 1: **Quantitative Evaluation.**

| | GraphDreamer | TC4D | HumanGaussian | Ours (Var-A) | Ours (Var-B) | Ours (static) | Ours |
|---|---|---|---|---|---|---|---|
| **CLIP-Image** ↑ | 98.44 | 89.50 | 83.93 | 97.88 | 92.11 | **93.45** | 92.20 |
| **CLIP-S** ↑ | 8.09 | 19.84 | 23.69 | 25.36 | 30.57 | 32.27 | **32.84** |
| **CLIP-DS** ↑ | 1.71 | 15.28 | 4.71 | 0.91 | 25.90 | **33.80** | 28.03 |

them, CLIP-S measures the similarity between texts and their corresponding models, CLIP-Image denotes the similarity between compositional models and human models, and CLIP-DS represents the alignment between changes in text captions (*e.g.*, "Iron Man" to "Iron Man holding an axe of Thor in his hand") and corresponding changes in images. Through Tab. 1, our method maintain the human identity in the composited scenes (see CLIP-Image). Note that "Ours (Var-A)" and GraphDreamer is slightly better for this metric as they struggle to do the composition (see Fig. 3). Meanwhile, "Ours" and "Ours (static)" consistently achieve better results than HumanGaussian and other variants, further affirming the objective superiority of AvatarGO.

**User Studies** We further conduct user studies to compare with DreamGaussian4D, HumanGaussian, TC4D, and "Ours (Var-A)". 24 Volunteers rated these methods independently based on seven criteria

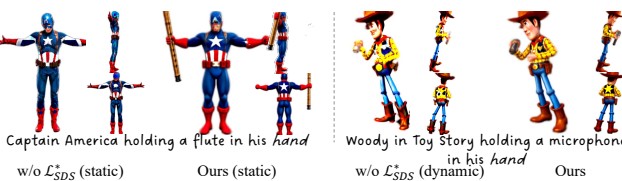

Figure 6: **Analysis of correspondence-aware motion field.**

from 1 (worst) to 5 (best): (1) Level of penetration; (2) Accuracy of the relative scale between humans and objects; (3) Accuracy of contact; (4) Motion quality; (5) Motion amount; (6) Text alignment; (7) Overall performance. Detailed results have been presented in Tab. 5. Key observations include: 1) Both DreamGaussian4D and HumanGaussian have difficulty providing satisfactory outcomes for human-object interaction (HOI) scenes. 2) Although TC4D performs well with HOI generations, it only produces global motions, leading to less optimal motion quality and quantity compared to our method. Our final design consistently delivers superior results for all seven criteria, outperforming the other methods across the board.

### 4.3 ABLATION STUDIES

**Analysis of LLM-guided Contact Retargeting.** We first conduct evaluations to validate the efficacy of employing Lang-SAM for retargeting the accurate contact area. See Fig. 3. By comparing "Ours (**Var-A**)" and Ours, we can conclude that without Lang-SAM, the model struggles to produce correct human-object interaction in the 3D compositional generation.

**Analysis of Correspondence-aware Motion Field.** In Fig. 6, we first compare our proposed training objectives $\mathcal{L}_{CA}$ with two alternative strategies: 1) "SDF distance loss" which minimizes the change of signed distance field (SDF) between objects and humans when they are animated to a new pose, and 2) "SDF label loss"

Figure 5: **User studies.**

| | Dream Gaussian4D | Human Gaussian | TC4D | Ours (Var-B) | Ours |
|---|---|---|---|---|---|
| Level of penetration ↑ | 1.267 | 1.084 | 4.236 | 1.537 | **4.872** |
| Accuracy of relative scale ↑ | 1.183 | 1.092 | 4.308 | 3.947 | **4.788** |
| Accuracy of contact ↑ | 1.654 | 1.137 | 4.412 | 2.137 | **4.802** |
| Motion quality ↑ | 1.321 | 2.156 | 1.947 | 1.673 | **4.592** |
| Motion amount ↑ | 2.118 | 3.781 | 1.517 | 4.159 | **4.934** |
| Text alignment ↑ | 2.047 | 1.918 | 4.515 | 2.462 | **4.767** |
| Overall Performance ↑ | 3.467 | 1.633 | 4.033 | 2.033 | **4.869** |

that supervise the label of SDF instead. These comparisons demonstrate the effectiveness of our proposed method for maintaining spatial correlations during the animations. Additionally, we validate our model's design by further comparing it with two variants: 1) "w/o $\mathcal{R}$, $\mathcal{T}$, $\mathcal{L}_{CA}$" which disables the trainable parameters $\mathcal{R}$, $\mathcal{T}$, Eq. 15, and our proposed loss $\mathcal{L}_{CA}$. This setting represents the scenario where the object is moved directly with the contact point. and 2) "w/o $\mathcal{L}_{CA}$" which trains the animation network solely with SDS loss ($\mathcal{L}_{SDS}^*$, $\mathcal{L}_{SDS}$). These comparisons underscore the necessity of these components in achieving 4D animation with better robustness to the penetration issues.

**Analysis of Spatial-aware SDS.** We finally assess the effectiveness of spatial-aware SDS (SSDS) and present the results in Fig. 7. Notably, we observe that SSDS plays a crucial role in preventing the optimization of $\mathcal{R}$, $\mathcal{S}$, $\mathcal{T}$ from vanishing during 3D compositional generation. Addition-

Captain America holding a flute in his *hand*

w/o $\mathcal{L}_{SDS}^*$ (static)    Ours (static)

Woody in Toy Story holding a microphone in his *hand*

w/o $\mathcal{L}_{SDS}^*$ (dynamic)    Ours

Figure 7: **Analysis of Spatial-aware SDS.**

ally, there is a drop in the quality of the animated avatars when disabling SSDS.

## 5 CONCLUSIONS

In this paper, we have introduced AvatarGO, the first attempt for text-guided 4D avatar generation with object interactions. Within AvatarGO, we proposed to employ large language model for comprehending the most suitable contact area between humans and objects. We also presented a novel correspondence-aware motion optimization that utilizes SMPL-X as an intermediary to enhance the model's resilience to penetration issues when animating 3D humans and objects into new poses. Extensive evaluations demonstrated that our method has achieved high-fidelity 4D animations across diverse 3D avatar-object pairs and poses, surpassing current state-of-the-arts by a large margin.

## ACKOWLEDGEMENT

This study is supported by the National Key R&D Program of China No.2022ZD0160102. This study is also supported by Shanghai AI Laboratory, the Ministry of Education, Singapore, under its MOE AcRF Tier 2 (MOE-T2EP20223-0002, MOE-T2EP20221-0012), and under the RIE2020 Industry Alignment Fund – Industry Collaboration Projects (IAF-ICP) Funding Initiative, as well as cash and in-kind contribution from the industry partner(s). This study is also supported by the Hong Kong Research Grants Council - General Research Fund (Grant No.: 17211024).

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
