# OpenReview forum: "AvatarGO: Zero-shot 4D Human-Object Interaction Generation and Animation"
_ICLR.cc/2025/Conference — ICLR 2025 Poster_

### Official Review · Reviewer_bzXT · 2024-11-01

**Soundness:** 3
**Presentation:** 3
**Contribution:** 3
**Rating:** 5
**Confidence:** 3

**Summary:**

This paper introduces AvatarGO, a novel framework for generating realistic and animatable 4D human-object interaction (HOI) scenes directly from textual descriptions. Addressing the challenges of determining "where" and "how" objects interact with human bodies, the authors propose two key innovations:

1. **LLM-Guided Contact Retargeting**: Utilizing Lang-SAM to identify contact body parts from text prompts, ensuring precise human-object spatial relations.
2. **Correspondence-Aware Motion Optimization**: Constructing motion fields for both human and object models using the linear blend skinning (LBS) function from SMPL-X to maintain correspondence during animation, reducing penetration issues.

The proposed method operates in a zero-shot manner using pre-trained diffusion models, overcoming the scarcity of realistic large-scale interaction data. Extensive experiments demonstrate that AvatarGO outperforms existing methods in generating coherent compositional motions and handling various human-object pairs and poses.

**Strengths:**

- The paper tackles the challenging and less-explored problem of generating visually realistic and animated 4D human-object interactions directly from textual inputs, which is crucial for applications in AR/VR and gaming.
- To the best of my knowledge, using LLM-Guided Contact Retargeting and Correspondence-Aware Motion Optimization to improve HOI in 4D Gaussian Splatting generation is novel. The proposed methods clearly solve the “where” and “how” problems raised in the introduction.
- The proposed method operates in a zero-shot manner with pre-trained diffusion models. It avoids the need for scarce large-scale interaction datasets.

**Weaknesses:**

A significant problem of the paper is the lack of a substantial body of related work in HOI generation, particularly in 4D HOI generation methods. Important works such as Programmable Motion Generation, Chain-of-Contacts, CG-HOI, HSI, among others, are not discussed. These methods have contributed significantly to the field by generating SMPL and SMPL-X poses for HOI scenarios.

While the paper focuses on generating realistic appearances using Gaussian splatting, which differs from the approaches of these methods, it is essential for the authors to analyze and position their work in the context of existing studies in 4D HOI generation. For instance, the use of large language models (LLMs) to generate contact-based constraints was first proposed in Chain-of-Contacts. By not acknowledging these relevant works, the paper misses the opportunity to highlight how AvatarGO builds upon or differs from established methods, potentially overlooking important insights and contributions from the existing literature.

Another weakness of the proposed method, as mentioned by the authors in the limitation section, is that it assumes a consistent geometric relationship between the interacting body part and the object. This prohibits it from generating complex manipulations such as tossing an object or passing an object between hands.

### References

- Liu, Hanchao, et al. "Programmable Motion Generation for Open-Set Motion Control Tasks." *Proceedings of the IEEE/CVF Conference on Computer Vision and Pattern Recognition (CVPR)*. 2024.
- Xiao, Zeqi, et al. "Unified Human-Scene Interaction via Prompted Chain-of-Contacts." *The Twelfth International Conference on Learning Representations (ICLR)*.
- Diller, Christian, and Angela Dai. "Cg-hoi: Contact-guided 3d human-object interaction generation." *Proceedings of the IEEE/CVF Conference on Computer Vision and Pattern Recognition*. 2024.
- Jiang, Nan, et al. "Scaling up dynamic human-scene interaction modeling." *Proceedings of the IEEE/CVF Conference on Computer Vision and Pattern Recognition*. 2024.

**Questions:**

- How does AvatarGo compare to SMPL-based 4D HOI generation methods?
- Aside from the apparent appearance advantage, does AvatarGo achieve a similar interaction quality to the SMPL-based methods?
- What advantages does direct 4D Gaussian splatting optimization offer over a two-stage approach (SMPL generation followed by GS fitting)?
- Why do the results trend toward anime/cartoon styles? What technical modifications would enable more photorealistic outputs?
- How does runtime scale with sequence length and scene complexity?

---

> ### Author Response · Authors · 2024-11-28
>
> # Response to reviewer `#bzXT` (1/2)
>
> We appreciate the reviewer's inquiry regarding the comparison with SMPL-based 4D HOI generation methods. We acknowledge the valuable contributions made by existing methods in advancing human motion generation.
>
> Our approach builds upon the strengths and robustness exhibited by these methods, aiming to enhance the realism and usability of 4D HOI generation. With a better 4D HOI method, our approach will also show improved performance. We hope for this paper to not only showcase the potential of 4D HOI but also to potentially advance and elevate our field.
>
> > a substantial body of related work in HOI generation, particularly in 4D HOI generation methods. These methods have contributed significantly to the field by generating SMPL and SMPL-X poses for HOI scenarios.
>
> > How does AvatarGo compare to SMPL-based 4D HOI generation methods?
>
> We concur with the reviewer's observation that Programmable Motion Generation, Chain-of-Contacts, CG-HOI, and HSI have significantly advanced the field. However, these advancements primarily focus on generating motion sequences for SMPL or SMPL-X models, which lack intricate clothing details. Additionally, these methods rely on training datasets, limiting their adaptability in practical scenarios. While InterDreamer recently introduced a zero-shot framework for generating appearance within 4D HOI, their models still closely resemble the original SMPL design, resulting in minimal clothing details in their outputs. In contrast, our approach offers a novel avenue for zero-shot text-guided 4D HOI generation, producing both realistic appearance and geometry enhancements.
>
> > While the paper focuses on generating realistic appearances using Gaussian splatting, which differs from the approaches of these methods, it is essential for the authors to analyze and position their work in the context of existing studies in 4D HOI generation.
>
> Similar to the above discussion, our approach presents a novel opportunity for zero-shot text-guided 4D HOI generation, emphasizing the generation of realistic appearance and geometry. We will include more discussion with these 4D HOI generation methods and highlight the importance of the SMPL-based 4D HOI methods within this research domain.
>
> > Another weakness of the proposed method, as mentioned by the authors in the limitation section, is that it assumes a consistent geometric relationship between the interacting body part and the object.
>
> We agree with the reviewer on this point. However, We believe our approach to handling rigid-body interactions and continuous contact in 4D human animation represents a significant advancement. Meanwhile, this scenario, where the generated avatar interacts with rigid objects, is common in real-world applications, yet existing methods struggle with it. In other words, our proposed method outperforms all existing related approaches. Moreover, similar applications in robotics, such as embodied AI, primarily focus on resolving interactions between rigid structures. Similarly, SAGA[1] and other works focus only on a small range of settings, such as approaching and grasping an object. As an early study on generating 4D HOI scenes, we believe our method is valuable to the research community and will inspire future studies addressing more challenging cases. We will also aim to solve this problem in the future works.
>
> > Aside from the apparent appearance advantage, does AvatarGo achieve a similar interaction quality to the SMPL-based methods?
>
> Our methodology is centered around continuous interactions between humans and objects, making direct comparisons with SAGA[1] and similar works that concentrate on object grasping somewhat unfair.
>
> When comparing with SMPL-based approaches operating within a similar context, we would like to direct the reviewer to the webpage of InterDreamer[2] (their code is not publicly released for comparison). It is noticeable that this method encounters challenges in maintaining realistic human-object interactions, often leading to insufficient contact or significant penetration. These observations demonstrate that our method achieves comparable or superior performance when compared to existing SMPL-based techniques.

---

> > ### Author Response · Authors · 2024-11-28
> >
> > # Response to reviewer `#bzXT` (2/2)
> >
> > > What advantages does direct 4D Gaussian splatting optimization offer over a two-stage approach (SMPL generation followed by GS fitting)?
> >
> > Our method aligns with this setting, as we will first acquire human motion sequences before generating the 4D HOI scenes. This is why we emphasize that SMPL-based 4D HOI methods form the foundation of our approach.
> > Additionally, there are alternative approaches that incorporate SMPL as the geometric prior, such as DreamAvatar [3], TADA [4], and DreamWaltz [5]. However, these techniques encounter challenges in achieving dynamic 4D generation. While TADA and DreamWaltz enable avatar animation, they accomplish this through post-processing using SMPL-based motion sequences.
> >
> > > Why do the results trend toward anime/cartoon styles? What technical modifications would enable more photorealistic outputs?
> >
> > The bias stemming from the pre-trained diffusion model is a notable factor contributing to this issue. Similar challenges can be noted in existing 3D generation methods like DreamAvatar [3], TADA [4], and DreamGaussian [6].
> > A possible strategy to overcome this limitation is presented by AvatarBooth [7]. Through the fine-tuning of personalized diffusion models with human images, AvatarBooth enables the generation of more photorealistic models, showcasing a pathway to address and mitigate these biases effectively. Following AvatarBooth, our method can also enable more photorealistic outputs by using a personalized diffusion model.
> >
> > > How does runtime scale with sequence length and scene complexity?
> >
> > In our method, the runtime would increase linearly in correlation with the sequence length and scene complexity, but it would require more GPU memory for training.
> >
> > [1] SAGA: Stochastic Whole-Body Grasping with Contact. ECCV 2022.
> >
> > [2] InterDreamer: Zero-Shot Text to 3D Dynamic Human-Object Interaction. NeurIPS 2024.
> >
> > [3] DreamAvatar: Text-and-Shape Guided 3D Human Avatar Generation via Diffusion Models. CVPR 2024.
> >
> > [4] TADA! Text to Animatable Digital Avatars. 3DV 2024.
> >
> > [5] DreamWaltz: Make a Scene with Complex 3D Animatable Avatars. NeurIPS 2023.
> >
> > [6] DreamGaussian: Generative Gaussian Splatting for Efficient 3D Content Creation. ICLR 2023.
> >
> > [7] AvatarBooth: High-Quality and Customizable 3D Human Avatar Generation. arXiv 2023.

---

### Official Review · Reviewer_E4gh · 2024-11-03

**Soundness:** 3
**Presentation:** 3
**Contribution:** 3
**Rating:** 8
**Confidence:** 3

**Summary:**

This paper introduces AvatarGO, a novel framework that utilizes diffusion models and language guidance to enable zero-shot generation of realistic, animatable 4D human-object interactions from text prompts.

**Strengths:**

1. The paper is well-written, with a clear motivation and an easy-to-follow presentation of the core idea. Related work is thoroughly covered, incorporating up-to-date research in the field.
2. Extensive experiments and ablation studies convincingly demonstrate the effectiveness of AvatarGO, showcasing appropriate human-object interactions and superior robustness against penetration issues.

**Weaknesses:**

I believe this paper is ready for publication, as I found no major weaknesses. My only question concerns the diversity of objects included in the text prompts used in the experiments. Additionally, how does your method perform across different types of objects?

**Questions:**

N/A

---

> ### Author Response · Authors · 2024-11-28
>
> # Response to reviewer `#E4gh`
>
>
> Thank you very much for recognizing our paper! We appreciate your efforts during the stages of review and rebuttal.
>
> > My only question concerns the diversity of objects included in the text prompts used in the experiments. Additionally, how does your method perform across different types of objects?
>
> In our experiments, we incorporated over 15 randomly selected types of objects, and our methods demonstrated strong performance across this diverse set of objects.

---

### Official Review · Reviewer_1bKd · 2024-11-03

**Soundness:** 2
**Presentation:** 2
**Contribution:** 1
**Rating:** 3
**Confidence:** 3

**Summary:**

The paper proposes AvatarGO, a framework that generates 4D human-object interaction (HOI) to address the challenges of the lack of realistic large-scale interaction datasets. The authors generate 4D HOI samples in a zero-shot manner by leveraging a pre-trained diffusion model (Stable Diffusion). To mitigate the diffusion model's lack of understanding of "where" and "how" objects interact with humans, the authors propose 1) LLM-guided contact retargeting, which identifies the contact body part from the text prompt, and 2) correspondence-aware motion optimization, which models human and object deformation fields using linear blend skinning function. The authors argue that their extensive experiments validate the proposed method's superior generation capabilities.

**Strengths:**

1. The qualitative results of the proposed method seem more plausible than the compared methods in Figure 3. Moreover, the authors conduct user studies to compare the proposed method with comparable methods in diverse criteria, e.g., penetration, motion quality, and overall performance.

2. The paper is well-structured, with clear sections from introduction to methodology and experiments. The authors provide detailed explanations about their task with various citations, which helps the reviewer to understand the problem the authors focused on.

**Weaknesses:**

1. The reviewer wonders if the human motion is given as input or is generated by the proposed method. In Figure 2 caption (L185), the authors explain that correspondence-aware motion optimization jointly optimizes human and object animation, but in L345, it is written that the motion sequence is given. The reviewer requests clear explanations of what are the inputs and outputs of the proposed pipeline.

2. The reviewer thinks that the proposed method's novelty is limited. The objective function of the proposed generative model is spatial-aware SDS (L276), that is introduced by ComboVerse (Chen et al., 2024b), not newly proposed by the authors. Moreover, the authors propose LLM-guided contact retargeting, which identifies 3D Gaussians by unprojecting 2D segmentation masks. This process operates with a heuristically pre-defined threshold $\alpha$, and the reviewer wonders if this heuristic threshold $\alpha$ applies to various contact body parts, e.g., fingers, feet, head, and legs.

3. Since the proposed method only handles the composition of generated 3D humans and objects, the overall quality is prone to be bounded by the 3D asset generation module. Then, if the 3D objects and humans are generated in poor quality, how can the authors mitigate the error propagation in the proposed method?

4. The qualitative results in Figure 3 only show the interaction with human hands. The reviewer requests more visualization results of diverse objects and body parts, such as the head, feet, and legs.

**Questions:**

1. In L310, the reviewer wonders how the authors define the threshold $\alpha$.

---

> ### Author Response · Authors · 2024-11-28
>
> # Response to reviewer `#1bKd`
>
> > The reviewer wonders if the human motion is given as input or is generated by the proposed method.
>
> Our approach has the capability to accept either a human motion sequence directly as input or utilize a text prompt as input to generate a human motion sequence through existing techniques.
>
> > The reviewer requests clear explanations of what are the inputs and outputs of the proposed pipeline.
>
> Our method is capable of accepting either text prompts alone or a combination of text prompts and motion sequences as input to generate 4D dynamic human-object interactions. When only provided with text prompts, our approach initiates by generating the SMPL-based motion sequence via existing techniques before proceeding through our pipeline for further processing and generation.
>
> > The objective function of the proposed generative model is spatial-aware SDS (L276), that is introduced by ComboVerse (Chen et al., 2024b), not newly proposed by the authors.
>
> We definitely agree that the spatial-aware SDS (SSDS) was first introduced by ComboVerse, and we do not attribute SSDS as our contribution.
>
> Unfortunately, simply applying SSDS for 3D compositional generation may not lead to good results in human-object interactions.
> In Section 3.2, our primary contribution then lies in leveraging a large language model to extract contact area details, thereby improving the robustness of 3D human-object interaction generation.
>
> > This process operates with a heuristically pre-defined threshold alpha, and the reviewer wonders if this heuristic threshold alpha applies to various contact body parts, e.g., fingers, feet, head, and legs.
>
> In our method, we apply the same threshold alpha to various contact body parts, such as fingers, feet, head, and legs.
>
> > Then, if the 3D objects and humans are generated in poor quality, how can the authors mitigate the error propagation in the proposed method?
>
> We appreciate the question raised by the reviewer. Firstly, it's important to emphasize that our methodology is primarily designed to address interactions between humans and objects.
>
> Furthermore, existing techniques demonstrate a strong ability to generate high-quality 3D models for both objects and humans, despite some occasional instances of failure.
>
> In addition, our method still optimizes the attributes of 3D Gaussians when composing the human and object. This optimization leads to a slight improvement in quality and aids in mitigating common issues like "hole artifacts," which are often encountered in 3D Gaussian-based generation processes.
>
> > The reviewer requests more visualization results of diverse objects and body parts, such as the head, feet, and legs.
>
> In Fig. 1, we have showcased results depicting interactions involving the head and feet, as seen in the examples "Harry Potter - bear head hat" and "Naruto in Naruto Series - football."
>
> While we aim to present additional cases involving interactions with various body parts, we recognize that interactions between humans and objects predominantly occur around the hands, feet, and head.
>
>
> > In L310, the reviewer wonders how the authors define the threshold alpha
>
> We follow GaussianEditor[1] to define the threshold alpha as 0.5.
>
> [1] GaussianEditor: Swift and Controllable 3D Editing with Gaussian Splatting. CVPR 2024

---

### Official Review · Reviewer_D2nt · 2024-11-04

**Soundness:** 3
**Presentation:** 3
**Contribution:** 2
**Rating:** 6
**Confidence:** 4

**Summary:**

The paper introduces AvatarGO, a framework for generating and animating 4D human-object interactions (HOI) in a zero-shot manner using pre-trained diffusion models. The main contributions of the paper include: (1) LLM-guided ****contact ****retargeting, which addresses the challenge of determining where objects should interact with the human body by using large language models (LLMs) to identify appropriate contact areas from text prompts, and (2) correspondence-aware motion optimization, which ensures coherent motion between humans and objects while preventing issues like object penetration during animation. AvatarGO generates realistic 4D HOI scenes directly from textual inputs, overcoming limitations in existing methods that rely on SMPL-based models. Qualitative and quantitative results show that the AvatarGO outperforms existing methods in terms of generating plausible human-object interaction animation.

**Strengths:**

- The paper tackles the new task of zero-shot generation of 4d human-object interaction (HOI).
- The proposed LLM-guided contact retargeting helps initialize the object in reasonable 3D locations to help generate plausible HOI.
- The correspondence-aware motion optimization helps prevent penetration and maintain plausible interactions throughout the animation.
- Visually, the method outperforms previous methods in the quality of generated HOI. The higher CLIP score and user preference also support this.

**Weaknesses:**

- The overall appearance quality of generated humans and objects is still limited. The results are often blurry and lack realism. There is also still human-object penetration (see “Naruto in Naruto Series stepping on a football under his foot” on the webpage).
- The notations in object animation and correspondence-aware motion optimization are poorly explained. What is G_c and G_o? How are they computed? The paper mentioned they are derived from x_ci and x_oi, but it’s unclear.
- It seems the method requires training for each human-object pair, which can be quite inefficient for large-scale generation. Is there anyway for the method to train the same character interacting with multiple objects?

**Questions:**

- Is each HOI trained with one motion only? Can it generalize to multiple motions?
- Fig. 4 only shows one frame of animation, which is not animation but reposing. Multiple frames need to be shown.

---

> ### Author Response · Authors · 2024-11-28
>
> # Response to reviewer `#D2nt`
>
> > The overall appearance quality of generated humans and objects is still limited. The results are often blurry and lack realism. There is also still human-object penetration (see “Naruto in Naruto Series stepping on a football under his foot” on the webpage).
>
> While we acknowledge the reviewer's feedback regarding the quality of the generated humans and objects, it's important to emphasize that our work represents the pioneering effort in zero-shot 4D human-object interaction generation. For the "Naruto-football" case, we want to highlight that the penetration is minor, and achieving fully realistic results without such minor overlaps can be challenging. Rest assured, we are dedicated to further enhancing the quality of our work in forthcoming endeavors.
>
> Similarly to TC4D, our method also can deliver higher quality results with extended training durations. However, it is notable that TC4D necessitates approximately 15 hours for training, which is nearly 20 times longer than the training duration required by our method.
>
>
> > The notations in object animation and correspondence-aware motion optimization are poorly explained. What is G_c and G_o?
>
> G_c and G_o represent the affine deformations utilized to transform SMPL vertices from the canonical space to the observation space.
>
> > How are they computed?
> > The paper mentioned they are derived from x_ci and x_oi, but it’s unclear.
>
> We determine G for each SMPL vertex by applying Equation (6). In our experiments, we derive G_{c_i} by finding the nearest vertex in the canonical SMPL model to x_{c_i}. Similarly, G_{o_i} is obtained by locating the closest vertex in the observed (posed) SMPL model to x_{o_i}.
>
> In the upcoming revised version, we will enhance the clarity of this explanation for better understanding.
>
> > It seems the method requires training for each human-object pair, which can be quite inefficient for large-scale generation. Is there anyway for the method to train the same character interacting with multiple objects?
>
> Certainly, similar to other zero-shot text-guided 3D/4D generation techniques, our method necessitates training for each human-object pairing. However, once we have generated the 3D composite human-object interactions, our approach can then be trained to animate these interactions into various poses.
>
> Training a model capable of enabling the same character to interact with multiple objects is challenging without the requisite training datasets. Each type of object presents unique interaction dynamics; for instance, the interactions between a human and a book would differ from those between a human and a chair.
>
> In light of this, we plan to explore training a model tailored for category-based human-object interactions in the future. This approach will allow us to train a single character to engage with diverse objects effectively in future iterations of our work.
>
>
> > Is each HOI trained with one motion only? Can it generalize to multiple motions?
>
> Each HOI can be trained to exhibit various motions. Nonetheless, achieving this would require training a correspondence-aware motion field specific to each motion sequence.
>
> > Fig. 4 only shows one frame of animation, which is not animation but reposing. Multiple frames need to be shown.
>
> Thanks for the suggestion. In the revised edition, we will update Figure 4 to display multiple frames.

---

### Author Response · Authors · 2024-12-01

We thank all reviewers for their time and effort in reviewing our paper!

Our method presents the first method for zero-shot text-guided 4D human-object interaction (HOI) generation featuring realistic geometry and texture. To realize this objective, our method makes two primary contributions:

**LLM-guided contact retargeting:** By utilizing Lang-SAM, we pinpoint the contact body part based on the text prompts, enhancing the precision of human-object spatial relationships.

**Correspondence-aware motion optimization:** Leveraging the linear blend skinning function from SMPL-X, we construct two distinct motion fields for humans and objects, improving the overall motion quality.

Being the first endeavor to synthesize 4D avatars engaged in object interactions, our method attains state-of-the-art performance in both composite 3D HOI generation and dynamic 4D HOI animation, demonstrating robustness against penetration issues.

**Below, we summarize the changes made according to the reviews:**

1. We discuss the factors contributing to the quality of our results, in terms of the geometry and appearance. Additionally, we present the potential of the generated quality (`#D2nt`).

2. We explain the meaning of the components in our correspondence-aware motion fields (Eq. (16)) and show the process for obtaining them (`#D2nt`).

3. We explain the setting of our method to generate the dynamic scene for each pair of humans and objects. we explore the potential for extending our method to enable a single human to interact with a range of objects and animate them with diverse motions without the need for re-training (`#D2nt`).

4. We will improve the Fig.4 to showcase multiple frames in the revised edition (`#D2nt`).

5. We discuss the inputs and outputs of our method (`#1bKd`).

6. We explain how we employ spatial-aware score distillation sampling while emphasizing the importance of our proposed LLM-guided contact retargeting (`#1bKd`).

7. We describe how we use the threshold alpha in our experiments and provide an explanation of how it is defined (`#1bKd`).

8. We showcase how the quality of the generated human and object impacts our work, such as when the quality is poor, and we explain how our method mitigates this influence. (`#1bKd`).

9. We showcase our method to generate diverse interaction between human and various body parts. Additionally, we explain the common parts where interactions between humans and objects typically occur (`#1bKd`).

10. We discuss the different objects in the text prompts we used for our experiments and how well our method works with various types of objects (`#E4gh`).

11. We further discuss our work with existing SMPL-based 4D HOI generation techniques and agree that the SMPL-based 4D HOI generation techniques lay the basis of our method (`#bzXT`).

12. We discuss the limitation in our current approach, specifically in addressing only continuous contact between humans and objects (`#bzXT`).

13. We present comparisons in terms of interaction quality with SMPL-based methods (`#bzXT`).

14. We discuss the potential and challenges of achieving our objectives in a two-stage process (`#bzXT`).

15. We explain why our method tends to produce a cartoon-like style and explore methods for attaining photorealistic results (`#bzXT`).

16. Lastly, we discuss how the runtime of our method scales with the length of the motion sequence and the complexity of the scene (`#bzXT`).


We sincerely thank all reviewers and the AC(s) again for their valuable suggestions, which have greatly helped strengthen our paper.

If you have any further questions, we would be happy to discuss them!

---

### Meta-Review · Area_Chair_U1Ek · 2024-12-23

**Metareview:**

This paper proposes a method for diffusion-guided synthesis of 4D human and object interactions. The key innovations that the work introduces to address this problem are: (a) LLM-guided contact retargeting: a language-guided feedback module that segments the Gaussians pertaining to the part of the body that interacts with the object and places the object at the start of the optimization process to learn a joint composition of the object and person, close to it; and (b) Correspondence-aware motion optimization: a correspondence-aware model for learning the joint animation of the human body and the object to explicitly avoid penetrations between them. This work is compared to several existing baselines to demonstrate its efficacy both quantitatively and qualitatively, along with several ablation studies with visual results only to showcase the contribution of each of the method's individual components.

**Additional Comments On Reviewer Discussion:**

Four reviewers provided scores of 6, 3, 8, 5. All reviewers noted the work to be the first zero-shot method to explore the joint generation of human and object interactions and their animation. However, the reviewers noted concerns with the overall visual quality and penetration (D2nt, 1bKd), lack of novelty because of the spatial SDS loss being introduced in prior work (1bKd), lack of results with multiple body parts (1bKd), clarity of the method (D2nt) and the lack of positioning and discussion of related work in HOI generation (bzXT). While the reviewers did not engage with authors during the rebuttal phase, the AC thoroughly read the paper, the authors' responses and the reviewers' feedback. The AC feels that the work is sufficiently novel, both in addressing a new problem and in proposing several innovative and technically sound solutions to address it, beyond SSDS. It is also sufficiently different from prior work in HOI generation, which only generate SMPL sequences and not actual images, which can model further things like cloth movement, etc. The AC also feels that the paper is well-written and the issues related to clarity are minor and easily addressable. Lastly, with respect to the overall visual quality of the results, it could indeed be improved further, whoever given that this can be directly improved in the future by incorporating human and object generation models with improved appearance, the AC feels that this is orthogonal to the main contribution of this work, which is around modeling better human and object interactions.

Hence, as the first work to even address this challenging problem, overall, while not perfect, the AC feels that this work advances the field forward w.r.t modeling to human and object interactions and recommends acceptance. The authors should incorporate the changes that they promised in their response to the reviewers into the final version of their paper.

---

### Decision · Program_Chairs · 2025-01-22

Accept (Poster)